# Modeling and Analysis of Human Comfort in Human–Robot Collaboration

**DOI:** 10.3390/biomimetics8060464

**Published:** 2023-10-01

**Authors:** Yuchen Yan, Haotian Su, Yunyi Jia

**Affiliations:** Department of Automotive Engineering, International Center for Automotive Research, Clemson University, Greenville, SC 29607, USA; yucheny@clemson.edu

**Keywords:** human comfort, human–robot collaboration, comfort modeling

## Abstract

The emergence and recent development of collaborative robots have introduced a safer and more efficient human–robot collaboration (HRC) manufacturing environment. Since the release of COBOTs, a great amount of research efforts have been focused on improving robot working efficiency, user safety, human intention detection, etc., while one significant factor—human comfort—has been frequently ignored. The comfort factor is critical to COBOT users due to its great impact on user acceptance. In previous studies, there is a lack of a mathematical-model-based approach to quantitatively describe and predict human comfort in HRC scenarios. Also, few studies have discussed the cases when multiple comfort factors take effect simultaneously. In this study, a multi-linear-regression-based general human comfort prediction model is proposed under human–robot collaboration scenarios, which is able to accurately predict the comfort levels of humans in multi-factor situations. The proposed method in this paper tackled these two gaps at the same time and also demonstrated the effectiveness of the approach with its high prediction accuracy. The overall average accuracy among all participants is 81.33%, while the overall maximum value is 88.94%, and the overall minimum value is 72.53%. The model uses subjective comfort rating feedback from human subjects as training and testing data. Experiments have been implemented, and the final results proved the effectiveness of the proposed approach in identifying human comfort levels in HRC.

## 1. Introduction

The applications of human–robot collaboration (HRC) have been growing fast and received huge attention in the research field due to the fast development of robotics technologies for collaborative robots (COBOTs) in recent years.

Human–robot collaboration (HRC) is known as “the state of a purposely designed robotic system and operator working in a collaborative workspace” [1]. It is an interdisciplinary field that focuses on the collaboration of humans and robots as they achieve shared goals [2]. In the past decade, robot manufacturers developed their own collaborative robots based on the concept of HRC, and released them into the market: ABB Yumi, UR3, Kuka, and IIWA. These robots, also known as COBOTs, provide prospective and great solutions to complex hybrid assembly tasks, especially in smart manufacturing contexts [3]. Through human–robot interaction, the tasks can be split between humans and robots based on their capabilities to leverage their unique advantages [4,5]. Despite the tremendous efforts from both academia and industry, the market share and industry-level applications of these collaborative robots (COBOTs) are still limited and have huge space for improvement. One of the major impacting factors is the comfort of humans in HRC, which is usually less emphasized in COBOT development but critical to user acceptance in HRC [6,7,8].

Although human comfort is often considered as a simple concept of common sense, it in fact is much more complicated than many people assume. One disappointing fact is that academia has not reached a consensus on a universal definition of comfort yet, therefore leaving a huge challenge to precisely evaluate human comfort level [9]. Some researchers perceived comfort as two discrete states: comfort presence and comfort absence, while some others held the contrasting opinion, which claims that comfort and discomfort are two opposites on a continuous scale, ranging from extreme discomfort through a neutral state to extreme comfort [10,11]. Some researchers also viewed comfort as an optimal state in which the person stops taking action to avoid discomfort [12]. Despite all the unsettled disputes in academia, several points of view are supported by most researchers: (1) comfort is subjectively determined by each individual’s personal nature; (2) comfort can be affected by a wide variety of factors from multiple natures, such as physical, physiological, or psychological; and (3) comfort is affected by one’s reaction to environmental stimuli [10]. These statements were also used as guidelines in our study.

In recent years, some researchers investigated how to evaluate and improve human comfort in HRC scenarios. Weitian et al. [13] proposed a computational approach to model and quantify human comfort during human–robot collaborative manufacturing. Ross et al. [14] found that human comfort has a direct and immediate influence on the collaboration quality between the robot and its human partner and is also a significant factor for the robot to be aware of. Jessi et al. [15] developed a method of evaluating how the invasion of personal space by a robot affects human comfort. Przemyslaw et al. [16] examined human response to motion-level robot adaptation to determine its effect on team fluency, human satisfaction, and perceived safety and comfort. Alami et al. [17] proposed a framework that allows the robot to select and perform its tasks based on the human partner’s presence, needs, and preferences. Ciccarelli et al. [18] proposed a system to improve human postural comfort by optimizing robot behavior.

However, the research above limits their comfort evaluation methods by merely using subjective ratings or simple statistical comparison approaches. Thus, the results of the papers above can only prove the qualitative or simple quantitative relationship between human comfort levels and the HRC factors. In addition, the approaches and experiment cases usually only tune one factor at a time, which is not applicable to most real-world HRC scenarios. Therefore, there is a lack of a descriptive mathematical model providing a detailed evaluation and description of human comfort in HRC scenarios [19].

Many research efforts have leveraged the advantage of comfort measurements by utilizing physiological signals, e.g., electroencephalography (EEG), electrodermal activity (EDA), and blood volume pulse (BVP), in a machine-learning-based model to analyze the general human comfort in HRC. Shan et al. [20] applied machine learning techniques in conjunction with passive EEG measurement to classify occupants’ real-time thermal comfort states. Performances of different machine learning techniques were compared, and methods to select linear continuous features for class interpolation were also explored. The classification results with the linear discriminant analysis classifier using the full-set features achieved an accuracy above 90%. Maaoui’s work [21] used two methods, support vector machine (SVM) and Fisher discriminant, to recognize human emotions of amusement, contentment, disgust, fear, neutral, and sadness with multiple physiological signals, e.g., BVP, EDA, and skin temperature (SKT). The recognition results for different types of emotions turned out to be excellent, with an accuracy around 92%. Klingner et al. has demonstrated the feasibility of measuring cognitive load by analyzing pupil size data [22]. Some other eye-movement metrics, such as saccade parameters, are also found to be influenced by psychological stresses [23,24].

Although many research efforts were made in comfort modeling using physiological-signal-based approaches, such a model is difficult to be interpreted and understood intuitively by humans due to machine learning’s black-box characteristic. Therefore, given all the research gaps above, there is a lack of a descriptive mathematical model that is explainable and easy to perceive and also capable of providing a detailed evaluation and description of human comfort in HRC scenarios. In this study, we not only successfully proposed such a model but also validated its effectiveness and high accuracy with 10 test subjects. A series of HRC tasks with five varying robot-motion factors were designed and used in the experiment. Human comfort is considered as a continuous scale, ranging from extreme discomfort through a neutral state to extreme comfort in our study. Likert scales were used to collect the experiment data for modeling and final result analysis. A post-experiment feedback session was also carried out to collect free comments and thoughts from the subjects regarding the causes of their comfort changes during the experiments. A total of 270 experiment scenarios were designed in a task pool, and then 60 tasks were randomly selected from the pool and used for each test subject. The data samples were split in ratios of 75% and 25%, respectively, and used as training and testing samples. For each subject, the model runs 10 times, with each time resampling the training and testing data. Eventually, we tested out the effectiveness of our developed model that implemented a multi-linear regression math model by training and testing the model with the data we collected. The overall average accuracy among all participants is 81.33%, while the overall maximum value is 88.94%, and the overall minimum value is 72.53%. In addition, corresponding factor analyses were conducted in the Results section.

## 2. Experiment and Data Acquisition

### 2.1. Experimental Platform Setup

The experiment platform is shown in Figure 1. An ABB Yumi robot is placed on one side of the experimental platform. The human subject sits on a height-adjustable chair on the other side of the experimental workbench and interacts and collaborates with the robot in manufacturing tasks. We choose the most common collaborative tasks in manufacturing in the study; i.e., robots deliver parts to human.

The Yumi robot is controlled by our built control system in ROS [25]. The higher-level Yumi motions for both arms are generated and executed in ROS. For example, there is one cube part placed on the right side of the workbench. Given a delivery task, e.g., delivering the cube to the human’s right hand, the ROS control system will first generate the action plan to pick up the cube, move the cube, and then deliver the cube. Since the focus of this study is not robot autonomy but human comfort, we have structured the working environment where all object positions are known. Based on the positions, the control system will generate the motions in terms of trajectories of the end effector, and then the Yumi motion controller is used to generate and execute joint motions to drive the robot to follow these trajectories.

The human wears marker gloves on his/her right hand so that the motions of his/her hands can be precisely tracked by a Vicon motion tracking system. The human can use his/her hands to trigger the robot part delivery motions. For example, when the right hand is raised, it triggers the robot to deliver a part to his right hand.

### 2.2. HRC Tasks Design

In our experiment, five factors were taken into consideration during the design process: final delivery distance, robot moving speed, final delivery height, robot arm approaching trajectory, and delivery pose. These five factors were selected because they were proved to have large impacts on human comfort in HRC collaborations from previous studies, and are also the most frequently apparent and investigated factors in HRC human-factor-related studies [13,14,15,16,19].

The entire experiment scenario pool is composed of 270 delivery task options, while each task uses a combination set of different factor levels. As shown in Table 1, the first two factors—final delivery distance and robot moving speed—each have 5 factor levels to choose from, respectively. Final delivery height and approach trajectory each have 3 factor levels, respectively, while the last factor—delivery pose—has only 2 options. The final delivery distance is specified as the horizontal distance measured from the robot tool center point (TCP) to the human upper body, while the final delivery height is defined as the vertical distance between the TCP and the experimental platform. The robot moving speed refers to the average linear speed of the robot TCP movements within each task. The approaching trajectory is differentiated by the style of the moving path of the robot TCP during delivering actions. The delivery pose is indicated by the spatial orientation of the robot’s end effector at the end of the delivery motion. The three options for the approaching trajectory are straight-line delivery, left-curvature path delivery, and right-curvature path delivery. The two options for the delivery pose are the flat pose and the vertical pose of the end effector.

All the factor-level options are shown in Table 1 below. Different factor values were chosen and merged into a factor set, which generates an experiment task among the experiment scenario pool. As we can see in Table 1, there are five factor columns, with each column representing one robot-motion factor. Take the ‘Distance’ column as an example: there are 5 different options—{25, 37.5, 50, 62.5, 75}. In every HRC task, we only pick one out of these 5 distance options and use it as the final delivery distance for the current task. Likewise, repeat this process for the other 4 factors. In such a way, we can create a combination set like this—{25, 0.1, 15, Straight, Flat}. In this study, multiple factors are tuned at the same time for each task. Instead of adopting all possible combinations from the five factors, certain combinations are excluded. When the delivery distance from human is equal to or larger than 50 cm, the moving path tends to be very short, and thus the approaching trajectory does not make much difference. In such cases, only the straight-path trajectory option will be used.

Eventually, 270 combination sets were created, and each factor set was used as the robot motion configuration setting in the ROS controller for a task. The aim of the task design is to induce human comfort responses under different human–robot interaction contexts. The five factors have been adopted and a sufficient number of combinations provide a comprehensive coverage of the scenarios that a human subject will potentially encounter in HRC tasks. Thus, the comfort prediction model can be proved to be universally applicable and extendable to a wide range of HRC scenarios.

### 2.3. Comfort Data Collection

As introduced in the introduction section, subjective feedback and ratings from human subjects are usually considered as the ground truth values in human-factor-related studies. In our study, we followed this rule and implemented a 5-point Likert scale questionnaire to collect the subjective comfort level ratings from the subjects. After completing each HRC case of the experiment, subjects were asked to report an integer-only comfort rating within the range of [1, 5], while a smaller number represents less comfort and a larger number represents higher comfort. Test subjects were told to neglect irrelevant factors or stimuli as much as they could during the experiment.

Although 270 experiment scenarios were created, only 60 tasks were randomly selected from the pool and used for each subject due to consideration of time. The sampling of the 60 tasks is not a simple random sampling but roughly based on the factor combination options, trying to maximize the chances of the subject experiencing different factor changes.

In this study, we also specifically designed another smaller set of experiment cases as the training session, which contains some of the most extreme parameter setups. The necessity of implementing the training session before the official experiment starts includes two critical reasons. First of all, human subjects have to be familiar with the general moving pattern of the robotic arms and the pose of the robot’s end effectors so that their comfort ratings will not be affected by these non-related factors. We used the linear-style movements from the ABB motion planning algorithm so that the pose of the end effectors and general moving patterns should remain the same throughout the entire experiment, thus making a negligible impact on subjects’ comfort ratings. Without such training, subjects could easily be affected by uncertainty and unexpected stimulus. The second critical reason to have the training session is that the extreme parameter scenarios let the subjects know where the rating boundaries are so that they can better decide what rating scores to give based on their subjective feelings. Without knowing the extreme conditions of the interactions, the feedback ratings will usually be heavily concentrated within the medium range or one of the extreme ends.

At last, feedback ratings collected above were used as the ground truth values in both the training and testing processes of the comfort model.

### 2.4. Experiment Procedure

The training session was carried out first before starting the official experiment. During this training session, the authors provided basic introductions to the subjects on the experimental platform, experiment objective, robot’s characteristics and behaviors, data for collection, rules to follow, safety precautions, etc. Then, height variation among subjects was considered as we changed the position of each participant using an adjustable chair to make sure they were in the same relative height level with respect to the robot. Next, subjects started the training tasks and interacted with the robots. Subjects were told to ignore irrelevant factors’ impact, such as sudden noise or light reflection. Subjects experienced some extreme scenarios in this stage as well. After completing at least ten training cases, subjects were asked if they were ready for the official experiment. If not, the training session would be repeated until the subjects were fully prepared.

There are 60 scenario tasks in total for the official experiment. A random sequence is generated for the task completion order. The entire official experiment takes roughly 30 min to finish, without any short breaks or stoppages. This ensures the rating criteria consistency from the subjects. According to the post-experiment feedback, none of our subjects claimed that they felt exhausted without a break after the experiment finished.

## 3. Analytical Comfort Model

The analytical model is based on a new theory proposed in this paper that a human subject’s general comfort during an HRC task can be described with primitive comfort reward and combined comfort reward.

Primitive comfort reward is denoted as r, and defined as a normalized score that the human rates for a primitive factor, which affects human comfort in robot actions. In our case, the primitive comfort reward will be the subject’s comfort feelings towards different robot speed levels or delivery distances. Combined comfort reward is denoted as R, and defined as a normalized overall score that the human rates for a set of primitive factors that affect human comfort in robot actions. Readers should note that the subjective comfort ratings provided by the subjects during the experiment in this study are considered as the ground truth values for the combined comfort rewards, not the primitive rewards.

The combined comfort rewards, which indicate the overall comfort feeling of a human, are represented in a linear regression form with the set of primitive rewards, as shown in Equation (Equation 1). For a given test subject and a given set of independent factors $f_{1}$, $f_{2}$*…*$f_{N}$ that affect human comfort, one set of primitive comfort rewards can be described as $r_{i}$ $=[r_{i1},r_{i2}$…$r_{iN}]$. If *M* HRC tasks are carried out, or *M* sets of subjective ratings are reported, the collected primitive comfort reward is r$=[r_{1},r_{2}$…$r_{M}{]}^{T}$; the combined comfort reward is R$=[R_{1},R_{2}$…$R_{M}{]}^{T}$. For each task and its comfort evaluation, the corresponding combined comfort reward is defined in Equation (Equation 1):(1)$$R\left(r_{i}\right)=\sum_{j=1}^{N}\alpha_{j}r_{ij}+\alpha_{0}$$
where R∈[0,1], r∈[0,1], i∈[1,M]. *N* is the number of independent factors, which, in our case, equals five. $\alpha_{j}$ is the weight of the corresponding factor $f_{j}$, while $\alpha_{0}$ is the bias and is set to be 1. The process of building this comfort prediction model is to solve an optimization problem and obtain the optimal weight factor set and primitive comfort reward set. Under the given set of factors, the format of the human comfort model is defined as
(2)$$C\left(r_{i}\right)=R\left(r_{i}\right)+z_{i}$$
(3)$$S.T.:\left\{\begin{array}{cc} C(r_{i})=1 & \mathrm{if}\;C(r_{i})>1\\ C(r_{i})=0 & \mathrm{if}\;C(r_{i})<0 \end{array}\right.$$
where $r_{i}$ is the primitive comfort reward, $R(r_{i})$ is the combined comfort reward, $C(r_{i})$ is the comfort of human, and $z_{i}∼N\left(\mu_{i},\delta^{2}\right)$ is the comfort noise from human self-report. The loss function is given below in Equation (Equation 4): (4)$$L\left(\alpha \right)=\frac{1}{M}\sum_{i=1}^{M}{\left(R_{i}-\alpha^{T}\tilde{r_{i}}\right)}^{2}$$
where $\tilde{r_{i}}=[1,r_{i1},r_{i2}$…$r_{iN}]$, $\alpha =[1,\alpha_{1},\alpha_{2}$…$\alpha_{N}]$, $R_{i}$ is the subject’s self-reported comfort value for sample *i*. The optimization objective is to obtain the minimum of the loss function (4) under the given constraints. The constraints are given by
(5)$$0\leq \alpha_{i}\leq 1$$
where $\alpha_{i}$ are the factor weights. In this study, we implemented the interior-point algorithm [26] to solve the optimal α and *r* for the problem. The optimization problem is solved in an iterative process. The algorithm updates the solution *x* with a new estimation x+Δx. The search direction is determined by the KKT condition. The corresponding Hessian matrices are calculated with the BFGS method. The algorithm keeps iterating until reaching the minimal accuracy tolerance or the maximal iteration count.

## 4. Experiment Results and Analysis

### 4.1. Model Prediction Accuracy Results

Totally, ten graduate engineering students with an average age of 27.7 and a standard deviation of 3.68 contributed in data collection. Each subject contributed 60 samples of comfort data corresponding to 60 HRC tasks, respectively. For each subject, the model runs 10 times, with each time resampling the training and testing data. The amount ratios of training samples and testing samples are 75% and 25%, respectively. Therefore, in each test run, the comfort model is trained with 45 data samples and tested with the other 15 samples.

The average, maximum, and minimum comfort level prediction accuracies of all 10 participants are given in Table 2 and Figure 2. The overall average accuracy among all participants is 81.33%, while the overall maximum value is 88.94%, and the overall minimum value is 72.53%. The highest average accuracy result is 87.13% from subject 5, while the lowest average accuracy is 76.59% from subject 2. The range between maximum and minimum values among 10 running results for each subject is around 4–10%. Such performance stability can also be verified from the variance results in Table 2. Subjects 1 and 4 have relatively larger variances of 10.84 and 7.39, respectively, while most of the other subjects’ results are below 6. The performance gap among 10 running results within the same subject originates from the data sampling phase. Most subjects have few extreme ratings (1 or 5) throughout the entire experiment. In some running results, when most extreme cases are divided into the testing set, the prediction accuracy will be impacted due to the lack of training.

According to Table 2, all the participants have average accuracy over 76%, while seven out of ten have average accuracy over 80%, and five out of ten are higher than 83%. There are six out of ten participants with maximum accuracies over 85%, while only three participants achieved minimum accuracy lower than 75%. In general, the analytical model has a satisfying performance and proves its capability in human comfort evaluation in a complex multi-factor HRC scenario.

### 4.2. Comfort Factor Analysis

After reviewing the general accuracy results, we further investigated the factor level rewards and corresponding results, as shown in Table 3 and Figure 3, Figure 4, Figure 5, Figure 6 and Figure 7.

The average comfort reward values of different factor levels are listed in Table 3 above. Corresponding box plots are also shown in Figure 3, Figure 4, Figure 5, Figure 6 and Figure 7. The cross marks in the box plots represent the mean values, while the straight lines inside the boxes represent the median values.

For delivery distance, the average comfort values first slightly rise and then sharply drop as delivery distance increases. According to test subjects’ post-feedback comments, most subjects claimed that their comfort feelings improved as the delivery distance became closer. However, when the distance is too close to the human, the potential collision risk starts dominating the comfort feelings, which has negative effects. This explains the comfort rewards trend in Figure 3. The comfort feelings induced in this study can be converted and extracted into some more general comfort categories—mental comfort and physical comfort. The mental comfort is affected by patience, perceived safety, robot motion predictability, and expectation, etc. The physical comfort is affected by the magnitude/range of body movement, the body postural comfort, etc. [27]. An extremely short distance would induce high mental stress or safety concerns but require much smaller physical effort, or higher physical comfort. In the 25 cm distance case, the mental comfort dominates over the physical comfort, thus causing a slight drop in the overall rating, but physical comfort still plays the main role in the overall ratings in the majority of other cases. The 37.5 cm-distance case creates a perfect balance between the mental comfort and physical comfort, thus yielding the highest average comfort reward. Before obtaining the results, the hypothesis of the authors is that the data variance of 25 cm distance would be the largest because people tend to have opposite reactions under certain extreme conditions. For example, some people enjoy driving fast, while some people get nervous in a fast-driving vehicle. However, it turns out that the 50 cm distance case yields the largest variance. The probable reason for this could be that 50 cm is a dividing boundary for adults’ arm extension distance. Some people can easily reach up to 50 cm, while others struggle to do so.

For robot moving speed, the average comfort reward significantly increases as speed increases at first, then gradually decreases. Based on most subjects’ feedback, the slow-moving speed makes them impatient, while an extremely fast-moving speed will make some subjects nervous. Two mental comfort factors take major effects in this case—patience and perceived safety. In low-speed motion cases, patience dominates the mental comforts, while, in high-speed motion cases, perceived safety significantly drops, which negatively affects the overall comfort ratings. The speed configurations 0.3 m/s and 0.4 m/s not only have the highest comfort rewards but also yield the lowest data variances. This indicates that subjects have highly overlapped preferences in this speed range. The final delivery height factor results are shown in Table 3 and Figure 5. The average comfort reward constantly decreases as the delivery height increases. The result is simple to understand and intuitive as well: the higher the delivery position is, the harder it is for human subjects to reach, which causes stronger discomfort. One result worth noting is that the 30 cm height case has the smallest variance, much smaller than the 15 cm and 45 cm settings. This can be explained by the huge variation in subjects’ body heights. The 15 cm height can be too low for tall subjects, and 45 cm is too high for short subjects, but almost no subjects consider 30 cm height to be too high or too low.

For the robot delivering trajectory factor, the straight path option yields the highest average comfort reward, while left and right curvature paths are slightly lower, indicating that most subjects prefer direct deliveries. Despite such comfort reward differences, it is not hard to notice that these differences are very small. The difference between maximum and minimum average rewards for this factor is only 0.074, much smaller than the corresponding value of delivery distance—0.784. The probable reason for this could be due to the simplicity of the delivery task in this study. Some subjects claim that curvature paths are not favorable to them because they cannot identify the final goal and the purpose of the robot, causing them anxiety. The straight path option is always clear and predictable, which barely causes discomfort. We can assume that, if there are multiple objects positioned along the robot’s moving path, the predictability of the robot’s intention will significantly decrease, which will also cause a large drop in subjects’ mental comfort. This can be further investigated and verified in the authors’ future research.

As for the last comfort factor, delivery pose, the results from Figure 7 indicate that flat pose and vertical pose have similar rewards. The flat pose has a relatively higher average reward because, when the robot gripper is in the flat pose, it becomes a clear signal that the robot has stopped moving and reached its final destination. The default robot gripper is in the vertical position in motion and preparing states; thus, there is still a potential risk of sudden movement from the robot when the robot stops in a vertical delivery pose. Such uncertain and anxious feeling is the major reason that the flat pose has a higher comfort reward according to several subjects’ post-experiment feedback.

### 4.3. Factor Weights Analysis

The average comfort factor weight results are listed in Table 4 and Figure 8. The delivery distance has the largest average weight of 0.312 in comfort evaluation, while the delivery trajectory obtains the smallest weight of 0.1272. The robot moving speed has the second largest weight of 0.2292, while delivery height and delivery pose take the third and fourth places, respectively, with weights of 0.1767 and 0.1546. The reason that delivery distance has the largest comfort factor weight is that it is the only factor that is impacted by both mental and physical comforts. Delivery distance setup affects subjects’ perceived safety, the magnitude of body movement, and the body postural comfort at the same time. The two extreme distance cases trigger huge comfort variations on most subjects. When the test subject tries to reach the farthest delivery distance, the subject has to lean towards the upper body and fully extend his/her arm at the same time to be able to reach the target cube. On the other hand, in the shortest delivery distance cases, several subjects even leaned backward to secure a comfortable distance with the robot gripper. The other four factors only affect the general comfort either mentally or physically; thus, it is simple to understand that extreme delivery height will not cause such a level of discomfort because subjects only need to lift their arms up and down by a relatively small distance without any upper body movement.

## 5. Conclusions

In this paper, a multi-linear-regression-based general human comfort prediction model is proposed under human–robot collaboration scenarios. Previous related studies mostly utilize the subjective ratings method and questionnaires to evaluate how human comfort varies as one robot factor changes, yet such methods are limited in predicting comfort in a real-world scenario where multiple factors take effect simultaneously. Also, there is a lack of a mathematical-model-based approach to quantitatively describe human comfort in HRC scenarios. The proposed method in this paper tackled these two gaps at the same time and also demonstrated the effectiveness of the approach with its high prediction accuracy. The overall average accuracy among all participants is 81.33%, while the highest individual average accuracy result is 87.13% from subject 5, while the lowest individual average accuracy is 76.59% from subject 2. The overall maximum value is 88.94%, and the overall minimum value is 72.53%. All participants have average accuracy over 76%, while seven out of ten have average accuracy over 80%, and five out of ten are higher than 83%. There are six out of ten participants with maximum accuracies over 85%, while only three participants achieved minimum accuracy lower than 75%. Detailed factor analyses are also conducted in the results section, and final delivery distance is found to be the most influential factor. The range between maximum and minimum values among 10 running results for each subject is around 4–10%.

For future work, a physiological-data-based comfort model, also developed by the authors, can be used to fuse with the analytical model to further improve the overall prediction accuracy. In addition, a larger number of subjects with a wider range of ages will be recruited for data collection in future studies.

## Figures and Tables

**Figure 1 biomimetics-08-00464-f001:**
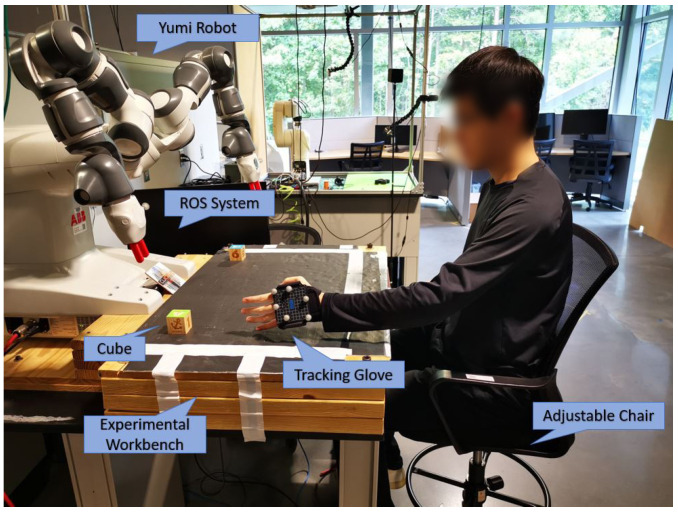
Human–robot collaboration platform with Vicon tracking system.

**Figure 2 biomimetics-08-00464-f002:**
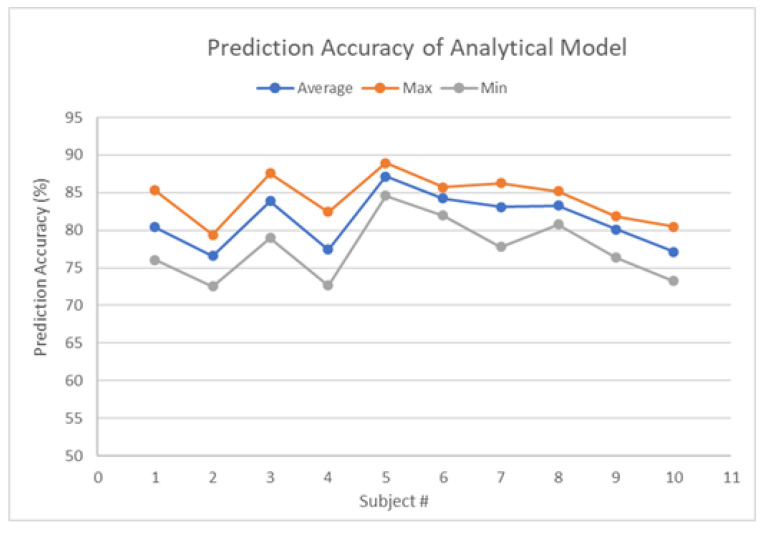
Prediction results of all participants with analytical comfort model.

**Figure 3 biomimetics-08-00464-f003:**
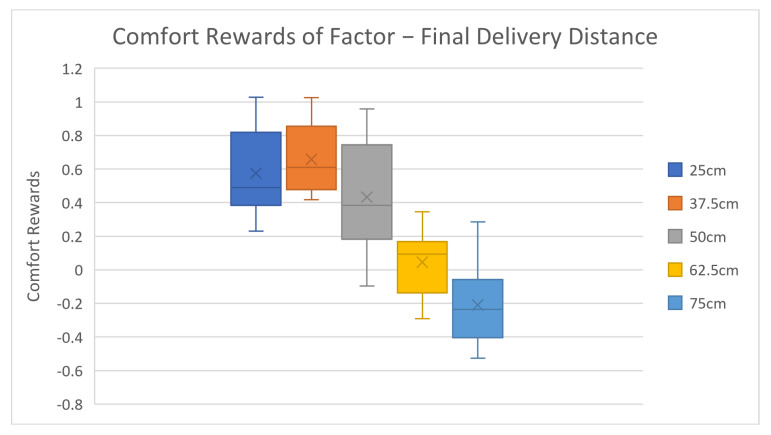
Average factor level rewards of final delivery distance.

**Figure 4 biomimetics-08-00464-f004:**
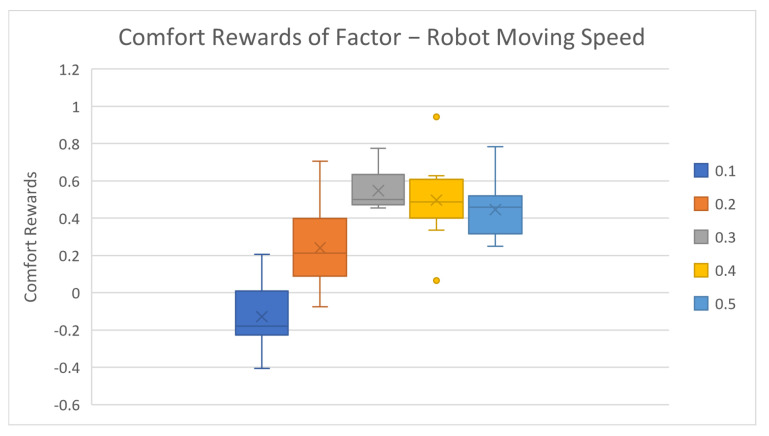
Average factor level rewards of robot moving speed.

**Figure 5 biomimetics-08-00464-f005:**
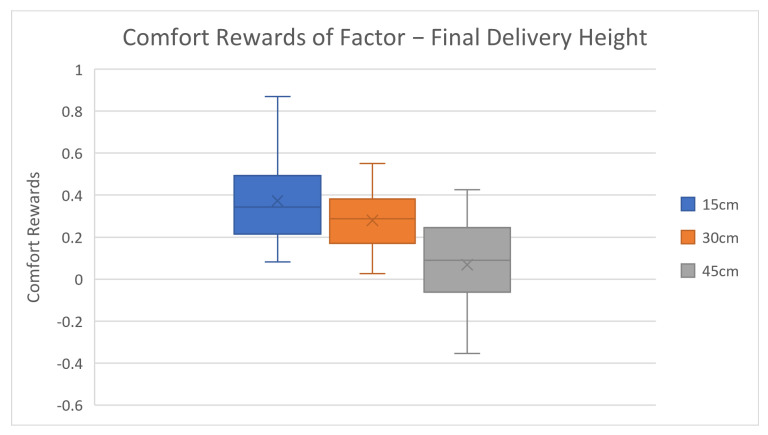
Average factor level rewards of final delivery height.

**Figure 6 biomimetics-08-00464-f006:**
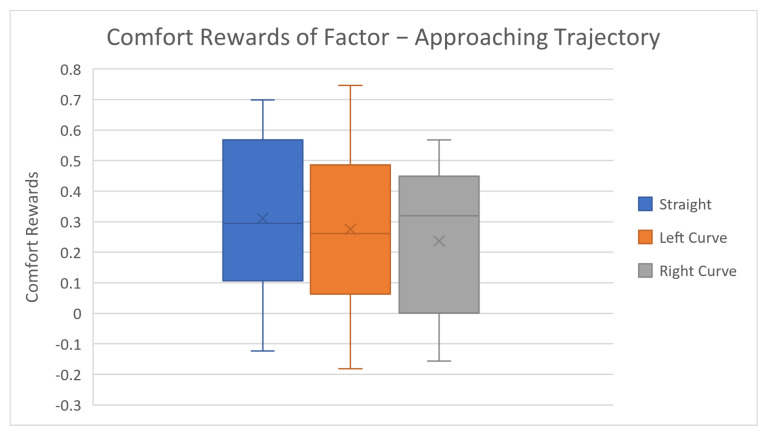
Average factor level rewards of approaching trajectory.

**Figure 7 biomimetics-08-00464-f007:**
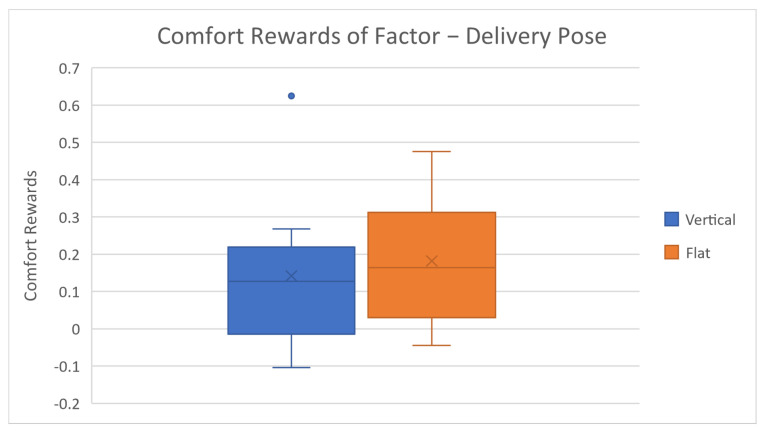
Average factor level rewards of delivery pose.

**Figure 8 biomimetics-08-00464-f008:**
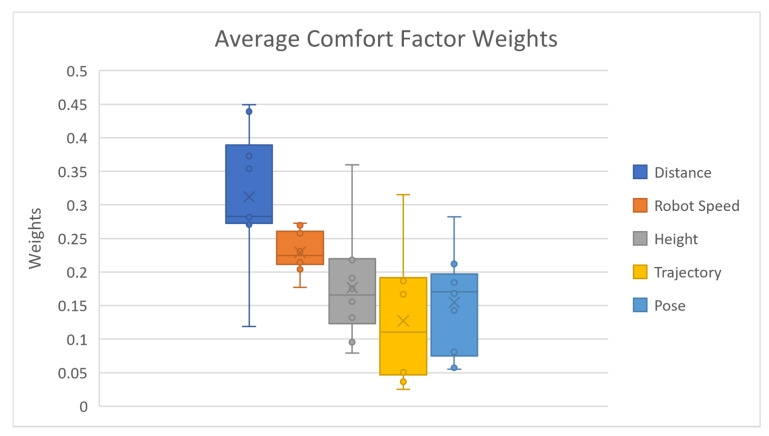
Average comfort factor weights.

**Table 1 biomimetics-08-00464-t001:** The factor combination set table.

Factors/Levels	Distance (cm)	Robot Speed (m/s)	Height (cm)	Approach Trajectory	Delivery Pose
1	25	0.1	15	Straight	Flat
2	37.5	0.2	30	Left Curve	Vertical
3	50	0.3	45	Right Curve	N/A
4	62.5	0.4	N/A	N/A	N/A
5	75	0.5	N/A	N/A	N/A

**Table 2 biomimetics-08-00464-t002:** Prediction accuracy of all participants.

	Average (%)	Max (%)	Min (%)	Variance
Subject 1	80.42	85.31	76.04	10.84
Subject 2	76.59	79.40	72.53	3.54
Subject 3	83.87	87.58	79.01	5.74
Subject 4	77.42	82.42	72.67	7.39
Subject 5	87.13	88.94	84.57	1.41
Subject 6	84.23	85.75	81.97	1.52
Subject 7	83.10	86.26	77.81	5.16
Subject 8	83.30	85.21	80.78	2.07
Subject 9	80.12	81.83	76.34	2.23
Subject 10	77.15	80.48	73.23	3.12
Overall	81.33	88.94	72.53	4.30

**Table 3 biomimetics-08-00464-t003:** Average factor level rewards of all factors.

Delivery Distance (cm)	Average Reward	Robot Speed (m/s)	Average Reward	Delivery Height (cm)	Average Reward	Trajectory	Average Reward	Delivery Pose	Average Reward
25	0.575	0.1	−0.127	15	0.373	Straight	0.310	Flat	0.182
37.5	0.658	0.2	0.240	30	0.279	Left Curve	0.275	Vertical	0.142
50	0.433	0.3	0.547	45	0.067	Right Curve	0.236	N/A	N/A
62.5	0.044	0.4	0.496	N/A	N/A	N/A	N/A	N/A	N/A
75	-0.209	0.5	0.446	N/A	N/A	N/A	N/A	N/A	N/A

**Table 4 biomimetics-08-00464-t004:** Average comfort factor weights.

Comfort Factors	Factor Weights
Distance	0.312
Robot Speed	0.2292
Height	0.1767
Trajectory	0.1272
Pose	0.1546

## Data Availability

Not applicable.

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
