# Peer review of "Modeling and Analysis of Human Comfort in Human–Robot Collaboration"

_biomimetics, 2023, doi:10.3390/biomimetics8060464_

Round 1
Reviewer 1 Report
BRIEF SUMMARY: The paper proposes a mathematical model-based approach to quantitatively describe and predict human comfort in HRC scenarios.
STRENGHTS: I think the topic of the manuscript is relevant for the field and the manuscript is presented in a well-structured manner.
WEAK ASPECTS AND RECOMMENDATIONS: The proposal has several weak aspects to be considered as a solid and practical piece of work:
- The introduction section should better establish which is the contribution of the paper. Moreover, it does not motivate sufficiently the problem that has been addressing in the paper and does not provide sufficient background and relevant references.
- Overall, Section 2 must be improved and give more detail of the decisions made. The reviewer recommends revise and expand all the section to make it more understandable. Provide more examples. For example in Section 2.2 authors take five factors into consideration, but why these five factors?
- Section 2.2 does not have the appropriate details to understand task design. Can authors provide examples of the HRC scenarios and HRC tasks? Also, in this section I don’t understand what a factor set is.
- In Section 3 authors should clearly define what is the primitive comfort reward and the combined comfort reward. At what theory refers the authors in line 199 (page 5)?
- In section 4.1 authors should give more detail about the simulations carried out.
- The conclusion section needs to be improved to provide more discussion of the results obtained and detail the further work.
- In Section 4, it should be blank spaces between “Table” and the number of the table, and between "Fig" and the number of figures.
The English language is appropriate and understandable, however some tenses are incorrect. In Section 2 there are some verbs in the future form that should be written in the present or past form. For example in page 3 (line 102), page 4 (line 157), page 5 (section 2.4).
Author Response
Major comments:
- The introduction section should better establish which is the contribution of the paper. Moreover, it does not motivate sufficiently the problem that has been addressing in the paper and does not provide sufficient background and relevant references.
Response:
Thank you for the valuable comment! We have revised the introduction to emphasize the motivation and contribution of the research with sufficient background and references. There are two major types of research gaps related to our study. Firstly, previous human comfort-related studies in HRC limit their comfort evaluation methods by merely using subjective ratings or simple statistical comparison approaches. Thus, the results of these studies tend to limit the analysis within the range of qualitative and simple quantitative relationships between the comfort factors and human comfort responses. Also, most studies usually only tune one factor at a time to reduce the complexity of the multi-factor effects. This is apparently not real in real-world HRC scenarios. Secondly, some research efforts were made in comfort modeling using physiological signal-based approaches, for example, using SVM to extract features from EEG, EDA, pupil signals and then classify the features into different comfort levels. However, this type of model cannot be easily interpreted by human.
Therefore, given all the research gaps above, there is a lack of a descriptive mathematical model which is explainable and easy to perceive, also capable of providing a detailed evaluation and description of human comfort in HRC scenarios. In this study, our contribution is that we not only successfully proposed such a model but also validated its effectiveness and high accuracy with 10 test subjects. The overall average accuracy among all participants is 81.33%, while the overall maximum value is 88.94%, and the overall minimum value is 72.53%.
We have added 5 new background references of previous studies and revised the contribution paragraph at the end of the introduction section. The revision can be found in Page 2 Paragraph 2-5.
- Overall, Section 2 must be improved and give more detail of the decisions made. The reviewer recommends revise and expand all the section to make it more understandable. Provide more examples. For example in Section 2.2 authors take five factors into consideration, but why these five factors?
Response:
Thank you for the valuable comment! In section 2.2, we introduced five factors which were taken into consideration for robot motion: final delivery distance, robot moving speed, final delivery height, robot arm approaching, trajectory and delivery pose. These five factors were chosen because they were proved to have large impacts on human comfort in HRC collaborations from previous studies. In addition, these factors are the most frequently appeared and investigated factors in HRC human factor-related studies. We have added some references of previous studies using these factors in Section 2.2.
The revision can be found in Page 4 Paragraph 1.
- Section 2.2 does not have the appropriate details to understand task design. Can authors provide examples of the HRC scenarios and HRC tasks? Also, in this section I don’t understand what a factor set is.
Response:
Thank you for the valuable comment! We have added more detailed descriptions of the interactive task using one combination set of factors as an example for better explanation in Section 2.2. A factor set is a set of robot motion parameters selected from Table 1 and then used in one specific HRC task. As we can see in Table 1, there are five columns, with each column representing one robot-motion factor. Take the ‘Distance’ column as an example, there are 5 different choices – {25, 37.5, 50, 62.5, 75}. In every HRC task, we only pick one out of these 5 distance choices and use it as the final delivery distance for the current task. Likewise, so on and so forth for other 4 factors. In such a way, we can create a combination set like this – {25, 0.1, 15, Straight, Flat}. In this set, the first number represents the delivery distance used in this task which is 25cm. The second number represents the robot’s moving speed which is 0.1m/s, similarly for other factors.
The revision can be found in Page 4 Paragraph 3.
- In Section 3 authors should clearly define what is the primitive comfort reward and the combined comfort reward. At what theory refers the authors in line 199 (page 5)?
Response:
Thank you for the valuable comment! We have given more clear definitions of the primitive comfort reward and combined comfort reward. In human-robot collaboration, the primitive comfort reward is a normalized score that the human rates for a primitive factor, which affects human comfort in robot actions. For example in our study, the primitive comfort reward can be the score for the factor level, final delivery distance – 25cm, in one experiment from one subject. In addition, the combined comfort reward is a normalized overall score that the human rates for a set of primitive factors, which affect human comfort in robot actions. For example in our study, the combined comfort reward can be the overall comfort score for an entire task in one experiment from one subject. As for the theory in line 199 (page 5), now in line 219 (page5), it is a new theory proposed in this paper to model human comfort in HRC and the model is validated through our experimental results in this paper.
The definitions can be found in Page 6 Paragraph 1.
- In section 4.1 authors should give more detail about the simulations carried out.
Response:
Thank you for the valuable comment! Since it is all based on real data, to make it easier to understand, we have changed the term “simulations” to “modeling results”. We have also expanded the description and explanation of the modeling results in section 4.1. The revised paragraph is rewritten as follows:
“Each subject contributed 60 samples of comfort data corresponding to 60 HRC tasks respectively. For each subject, the simulation runs 10 times, with each time resampling the training and testing data. The amount ratio of training samples and testing samples are 75% and 25%, respectively. Therefore, for each test run, the comfort model is trained with 45 data samples and tested with the other 15 samples.”
The revisions can be found in Page 7 Paragraph 1.
- The conclusion section needs to be improved to provide more discussion of the results obtained and detail the further work.
Response:
Thank you for the valuable comment! We have expanded the discussion of the results and further work details in the conclusion section. For the result discussion, we have included more detailed numbers in the conclusion section. As for the future work, we plan to recruit a larger number of subjects with a wider range of ages for data collection. The future experimental design will also account for variations in participants’ heights to make the model more accurate and adaptive to a larger human group.
The revisions can be found in Page 12 Paragraph 1-2.
- In Section 4, it should be blank spaces between “Table” and the number of the table, and between "Fig" and the number of figures.
Response:
Thank you for the valuable comment! We have added spaces in the revised version.
Minor comments:
The English language is appropriate and understandable, however some tenses are incorrect. In Section 2 there are some verbs in the future form that should be written in the present or past form. For example in page 3 (line 102), page 4 (line 157), page 5 (section 2.4).
Response:
Thank you for the valuable comment! We have revised and resolved the grammatical issues.
The revision can be found in Page 3 Paragraph 2, Page 5 Paragraph 1, Page 5 Paragraph 5-6.
Reviewer 2 Report
1 Previous studies in the field have primarily relied on subjective ratings and questionnaires to assess the impact of individual robot factors on human comfort. However, these methods have limitations when it comes to predicting comfort in real-world scenarios where multiple factors simultaneously come into play. To address this limitation, this paper introduces a novel approach—a multi-linear regression-based model for predicting human comfort in the context of human-robot collaboration scenarios.
2 In this study, a highly accurate predictive model for assessing comfort in Human-Robot Collaboration (HRC) tasks involving five factors—final delivery distance, robot moving speed, final delivery height, robot arm approaching trajectory, and delivery pose—has been developed. The model achieved an impressive maximum accuracy rate of 88.94% across all participants. Furthermore, the analysis of the results highlights that the final delivery distance exerts the most significant influence among these factors. These experimental findings hold valuable reference implications for future research in this domain.
3 This study conducted an experiment involving 10 engineering students; however, it's important to note that in real-world production settings, there may be a wider age range among workers. Additionally, the experimental design did not account for variations in participants' heights, which is an aspect that warrants further investigation in future research.
Author Response
Response:
Thank you for the valuable comments! The proposed comfort model is an individual-based model rather than a general comfort model for all. Therefore, ten subjects should be enough to validate the effectiveness of the modeling approach as they have different comfort models. However, we agree that collecting data from a wider range of ages for the study would further strengthen impact of the work. This has been discussed in the conclusion and future work. Regarding the heights, the height variation among subjects has actually been considered as we changed the position of each participant using a height-adjustable chair to make sure they were in the same relative height level with respect to the robot.
The revision can be found in Page 5 Paragraph 5, Page 12 Paragraph 2, highlighted in blue in the new manuscript.